# Modulatory Impact of Adipose-Derived Mesenchymal Stem Cells of Ankylosing Spondylitis Patients on T Helper Cell Differentiation

**DOI:** 10.3390/cells10020280

**Published:** 2021-01-30

**Authors:** Ewa Kuca-Warnawin, Iwona Janicka, Krzysztof Bonek, Ewa Kontny

**Affiliations:** 1Department of Pathophysiology and Immunology, National Institute of Geriatrics, Rheumatology, and Rehabilitation, 02-637 Warsaw, Poland; zaklad.patofizjologii@spartanska.pl (I.J.); ewa.kontny@spartanska.pl (E.K.); 2Department of Rheumatology, National Institute of Geriatrics, Rheumatology, and Rehabilitation, 02-637 Warsaw, Poland; krzysztof.bonek@gmail.com

**Keywords:** ankylosing spondylitis, adipose-derived mesenchymal stem cells, T helper cell subsets, transcription factors, cytokines

## Abstract

The domination of pro-inflammatory Th subsets (Th1, Th17) is characteristic of ankylosing spondylitis (AS). Mesenchymal stem cells (MSC) were reported to normalize Th imbalance, but whether MSCs from AS adipose tissue (AS/ASCs) possess such properties is unknown. We examined AS/ASCs’ impact on Th-cell differentiation, using healthy donors ASCs (HD/ASCs) as a control. The assessment of the expression of transcription factors defining Th1 (T-bet), Th2 (GATA3), Th17 (RORc), and Treg (FoxP3) subsets by quantitative RT-PCR, the concentrations of subset-specific cytokines by ELISA, and Treg (CD4^+^CD25^high^FoxP3^+^) formation by flow cytometry, were performed in the co-cultures of ASCs with activated CD4^+^ T cells or peripheral blood mononuclear cells (PBMCs). AS/ASCs and HD/ASCs exerted similar immunomodulatory effects. Acting directly on CD4^+^ T cells, ASCs decreased the T-bet/GATA3 and RORc/FoxP3 ratios, diminished Treg formation, but increase IFNγ and IL-17AF production, while ASCs co-cultured with PBMCs enhanced Treg generation and reduced IFNγ release. ASCs failed to up-regulate the anti-inflammatory IL-10 and TGFβ. AS/ASCs’ impact on allogeneic and autologous PBMCs was similar. In conclusion, to shift Th differentiation to a functional anti-inflammatory direction, ASCs require accessory cell support, whereas their direct effect may be pro-inflammatory. Because ASCs neither inhibit IL-17AF nor up-regulate anti-inflammatory cytokines, their usefulness for AS patients’ treatment remains uncertain.

## 1. Introduction

Ankylosing spondylitis (AS), a prototypical representative of spondyloarthritis, is characterized by chronic inflammation in spinal joints that leads to severe, persistent pain and pathological new bone formation with resulting spinal segments fusion. Slightly more patients with AS are male than female, and the approximate male to female ratio is 2–3:1. Peripheral arthritis, enthesitis, osteoporosis, and extra-skeletal symptoms are common, pointing out that AS is a systemic illness [1]. The disease pathogenesis is not fully understood, but the striking association with alleles encoding Human Leukocyte Antigen HLA-B*27 and endoplasmic reticulum aminopeptidases (ERAP) indicates that antigen presentation to T cells, followed by development and persistence of adaptive immune response, plays an important role [2]. The innate immunity cells contribute to AS pathogenesis as well [3,4,5], and the interleukin (IL-) 23/IL-17 cytokine axis, which affects both innate and adaptive immune cells, is thought to play a critical role. The circulating pool of IL-17 is supplied by several cell types, including T helper (Th) cells named Th17 [4,5,6]. The abnormalities of acquired immunity observed in AS patients concern mainly T lymphocytes and manifest as the T cell subsets imbalance [7,8,9], increased numbers of circulating Th17 cells [9,10], and defects of classical regulatory T (Treg) cells, which is thought to contribute to improper regulation of the immune response [11,12]. 

Mesenchymal stromal/stem cells (MSC), present in various tissues, exert immunosuppressive effects on different immune cells, including T lymphocytes [13,14]. Acting via soluble mediators and cell contact-dependent pathways, MSCs inhibit activation and proliferation of T cells, and directly or indirectly suppress disease-associated T cell subsets (Th1, Th2, Th17), whereas protect or induce Treg subpopulation [14]. Regulatory functions of MSCs in AS are relatively unknown. As bone marrow-derived, MSCs (BM-MSC) show abnormalities in gene expression, secretory potential, and immunomodulatory function [15,16,17], in currently ongoing clinical trials allogeneic MSCs are used for AS patients’ treatment [18]. Nevertheless, a case report of the beneficial application of autologous adipose-derived mesenchymal stem cells (ASCs) [19], and our recent reports showing normal phenotype and some basic biological activities of ASCs obtained from AS patients (AS/ASCs) [20,21], suggest therapeutic use of these cells. Therefore, in the present study, we have focused on further characterization of the immunoregulatory activity of AS/ASCs by investigating their impact on Th cell differentiation, using ASCs obtained from healthy donors (HD/ASC) as the reference cell lines. To this aim, the impact of ASCs on the expression of lineage-defining transcription factors of Th1, Th2, Th17, and Treg subsets as well as on the formation of classical (CD4^+^CD25^high^FoxP3^+^) Treg cells, and secretion of cytokines associated with respective Th subsets were investigated. To evaluate both the direct impact of ASCs on Th cell differentiation and the possible contribution of other cells in this process, ASCs were co-cultured with allogeneic purified CD4^+^ T cells and peripheral blood mononuclear cells (PBMCs), respectively. Finally, the sensitivity of patients’ target cells to autologous ASCs immunomodulation was assessed.

## 2. Materials and Methods

### 2.1. Patients and Ethics Approval

A group of 21 patients (9 female, 12 male) who fulfilled the Assessment of SpondyloArthritis International Society (ASAS) criteria for AS [22] were included in the study. Specimens of subcutaneous abdominal fat (approximately 300 mg) were taken from patients screened for amyloidosis, using a needle biopsy. Patients’ characteristics are given in Table 1 as well as in Appendix A. This study meets all criteria contained in the Declaration of Helsinki and was approved by the Ethics Committee of the National Institute of Geriatrics, Rheumatology, and Rehabilitation, Warsaw, Poland (the approval protocol no: KBT-8/4/20016). All patients gave their written informed consent before enrolment. 

### 2.2. ASCs Isolation and Culture

Subcutaneous abdominal fat tissue procurement from AS patients, tissue processing, and isolation of ASCs were performed according to the previously described protocol [23]. The five earlier characterized [20] human ASCs lines (Lonza Group, Inc., Walkershille, MD, USA) were used as a reference. All experiments were performed using ASCs at 3–5 passages. ASCs were cultured in a complete culture medium composed of Dulbecco’s Modified Eagle’s Medium/ Nutrient Mixture F-12 (DMEM/F12) (PAN Biotech UK Ltd., Wimborne, UK), 10% fetal calf serum (FCS) (Biochrom, Berlin, Germany), 200 U/mL penicillin, 200 µg/mL streptomycin (Polfa Tarchomin S.A., Warsaw, Poland) and 5 µg/mL plasmocin (InvivoGen, San Diego, CA, USA). For some experiments, ASCs were stimulated for 24 h with human recombinant tumor necrosis factor (TNF) and interferon γ (IFNγ) (both from R&D Systems, Minneapolis, MN, USA; each applied at 10 ng/mL).

### 2.3. Cell Co-Cultures

All cell co-cultures were performed in the complete culture medium (see above). ASCs (5 × 10^4^/well/2 mL of medium) were seeded into 24-well plates, stimulated with IFNγ and TNF (ASCs_TI_), or were left untreated. PBMCs were isolated from buffy coats obtained from healthy male donors (<60 years old), using Ficoll–Paque (GE Healthcare, Uppsala, Sweden) and routinely applied procedure. The CD3^+^CD4^+^ cells were isolated from PBMCs using EasySep™ Human CD4^+^ T Cell Isolation Kit (Stemcell Technologies, Vancouver, Canada). After isolation, PBMCs or CD4^+^ T cells (1.2 × 10^6^/well/2 mL of medium) were seeded either directly (contacting co-cultures) or on a 0.4 µm pore size Transwell filters (MD24 with a carrier for inserts 0.4 MY, Thermo Fisher Scientific, Massachusetts, MA, USA) (non-contacting co-culture) into 24-well plates with adherent ASCs (5 × 10^4^/well/2 mL of medium). Then, PBMCs were treated with 2.5 µg/mL of phytohaemagglutinin (PHA, Sigma–Aldrich, St. Louis, MO, USA), whereas CD4^+^ T cells were activated with Dynabeads™ Human T-Activator CD3/CD28 (Thermo Fisher Scientific, Massachusetts, MA, USA). After 5 days of co-culture, PBMCs or CD4^+^ T cells were harvested for cytometric analysis or RNA isolation. The concentrations of soluble factors were measured in collected culture supernatants. Separately cultured ASCs, PBMCs, and CD4^+^ T cells were used as the co-culture controls.

### 2.4. Flow Cytometry Analysis

For the CD4^+^ cells or PBMCs analysis, harvested cells were re-suspended in 50 μL of cell sorting buffer and stained for 30 min on ice for respective membrane antigens, using fluorochrome-conjugated monoclonal antibodies specific for human CD4–FITC, and CD25-PE (both from (BD Pharmingen, San Diego, CA, USA). In the next step, cells were fixed and permeabilized using FoxP3 transcription factor Staining Buffer Set (Thermo Fisher Scientific, MA, USA). Subsequently, intracellular staining using anti-FoxP3-APC antibody was performed. After the washing step, cells were acquired and analyzed using a FACS Canto cell cytometer and Diva software. Appropriate isotype controls were used in all experiments.

### 2.5. Enzyme-Linked Immunosorbent Assays (ELISA)

The concentrations of cytokines were measured in culture supernatants in duplicates by specific ELISAs. IL-17AF heterodimers, IFNγ, TGFβ, and IL-10 concentration measurements were done using Ready-SET-Go kits (eBioscience, San Diego, CA, USA).

### 2.6. Polymerase Chain Reaction (PCR)

RNeasy Mini Kit was used for mRNA isolation according to the manufacturer’s protocol (Qiagen, Hilden, Germany). High-Capacity cDNA Reverse Transcription Kit was used to reverse transcribe RNA to cDNA (Thermo Fisher Scientific, Waltham, MA, USA). The 10 µL PCR reaction included 2 µL RT product, 5 µL TaqMan Gene expression master Mix, 0.5 µL probe mix of the TaqMan (respectively, T-bet, GATA, RORc, FoxP3, GAPDH, TBP, RPL13a—all from Thermo Fisher Scientific, Waltham, MA, USA), and 2.5 µL of water (Genoplast, Poland). Reactions were performed at 50 °C for 2 min, 95 °C for 10 min, followed by 50 cycles at 95 °C for 15 s and 60 °C for 1 min. Samples were analyzed in triplicate using the QuantStudio 5 qRT-PCR machines (Thermo Fisher Scientific, Waltham, MA, USA). Gene expression was evaluated using ΔΔCT-method.

### 2.7. Statistical Analysis

Data were analyzed using GraphPad PRISM 5 software. One-way analysis of variance (ANOVA) with repeated measures and posthoc Tukey test was used to assess the effect of untreated and TNF/IFNγ-treated ASCs on target cells, and to compare cell contacting and non-contacting co-cultures. The Mann–Whitney test was applied to analyze differences between HD/ASCs and AS/ASCs. Probability values less than 0.05 were considered significant.

## 3. Results

### 3.1. Patients

The patients had clinically active AS, but moderate functional limitation. All patients were HLA-B27 positive, 45% of them presented ocular symptoms (uveitis), and 10% had peripheral arthritis. The majority of them were treated with non-steroid anti-inflammatory drugs (NSAIDs), whereas application of non-biologic disease-modifying anti-rheumatic drugs (DMARDs) and glucocorticoids was less frequent (Table 1). In our patients’ cohort there was no significant difference in clinical and demographic data between males and females, except the higher level of a systemic inflammation marker, C-reactive protein (CRP), in the female group, and a little more often treatment of females with DMARDs and glucocorticosteroids (Appendix A). The cytokine production and Treg generation were similar in the co-cultures of target cells with ASCs obtained from female and male patients (Appendix A) Therefore, in further analysis, the patients’ sex was not taken into account. 

### 3.2. ASCs Decrease T-Bet/GATA3 and RORc/FoxP3 Ratio in Activated CD4^+^ T Cells

As shown in Figure 1, in the presence of TI-treated and untreated HD/ASCs and AS/ASCs there was significant up-regulation of the expression levels of mRNAs encoding GATA3, RORc, and FoxP3 transcription factors in co-cultured, purified CD4^+^ T lymphocytes activated via CD3/CD28 pathway. By contrast, neither HD/ASCs nor AS/ASCs changed significantly the levels of T-bet mRNA in this experimental setting, and there was no difference between co-cultures containing untreated and TI-treated ASCs. As a result of the above modification of tested mRNAs expression, a significant decrease in T-bet/GATA3 ratio was observed. As for RORc/FoxP3 ratio, similar diminution was found in the co-cultures containing AS/ASCsTI, but in the presence of HD/ASCs opposite tendency was noted.

Similar changes in the expression of mRNAs coding for the above transcription factors were found in CD4^+^ T cells, when ASCs were co-cultured with PHA-activated PBMCs, i.e., decrease in T-bet/GATA3 mRNAs ratio due to up-regulation of GATA3 and relatively constant T-bet mRNA levels, as well as lowering of RORc/FoxP3 ratio despite significant up-regulation of both RORc and FoxP3 coding mRNAs (Figure 2). Although there were some small differences between effects exerted by HD/ASCs and AS/ASCs, they did not reach statistical significance, and a similar trend in tested mRNAs expression occurred in co-cultures of CD4^+^ T cells or PBMCs with both types of ASCs (Figure 1 and Figure 2). 

### 3.3. The Effects of ASCs on the Release of Th1 and Th17 Specific Cytokines

Neither HD/ASCs nor AS/ASCs secreted IL-17AF and IFNγ when cultured alone (data not shown). In the co-cultures of purified, activated CD4^+^ T cells with ASCs there was some increase in IFNγ and IL-17AF concentrations (Figure 3A,B), which in the case of IL-17AF was lower in the presence of AS/ASCs than HD/ASCs (Figure 3B). There was no difference between untreated and TI-treated ASCs. The results of these parts of experiments have shown that both types of ASCs up-regulate the generation of Th17 cells (RORc expression and IL-17AF production) directly. 

### 3.4. ASCs Modulate Generation of Classical Treg, But Fail to Up-Regulate Anti-Inflammatory Cytokines (IL-10 and TGFβ)

Comparing with separately cultured CD4^+^ T cells activated via CD3/CD28 pathway, the generation of classical Treg cells (CD4^+^CD25^high^FoxP3^+^) was reduced when CD4^+^ T cells were co-cultured with both untreated and TI-pre-stimulated HD/ASCs and AS/ASCs (Figure 4A). By contrast, there was a significant increase in the percentage of classical Treg cells when untreated and TI-pre-stimulated HD/ASCs and AS/ASCs were added to the co-cultures of PHA-activated PBMCs, as compared to activated PBMCs alone (Figure 4D). 

Separately cultured PHA-activated PBMCs release significantly less IL-10 than anti-CD3/CD28 activated CD4^+^ T cells (mean ± SEM for T vs. PBMCs, 2203 ± 290 vs. 552 ± 30 pg/mL, *p* < 0.0001), but secreted more TGFβ (mean ± SEM for T vs. PBMCs, 57 ± 14 vs. 520 ± 55 pg/mL, *p* < 0.0001). Separately cultured HD/ASCs and AS/ASCs did not secrete IL-10 at all and release small amounts of TGFβ (mean ± SEM = 145 ± 57 and 49 ± 22 pg/mL, respectively) (data not shown). Neither the presence of untreated nor TI-pre-stimulated HD/ASCs and AS/ASCs enhanced secretion of these anti-inflammatory cytokines that are associated with Tr1 type of regulatory T cells (Figure 4B–F). On the contrary, in PBMCs-ASCs co-cultures, a significant reduction in IL-10, and in the case of AS/ASCs TI also decrease TGFβ, was found (Figure 4E,F).

### 3.5. Contribution of the Cell-to-Cell Contact and Soluble Factors to ASCs Activity

As shown in Figure 5A–F, in the cell contacting co-cultures of HD/ASCs and AS/ASCs with PHA-activated PBMCs the secretion of IFNγ was significantly decreased while IL-17AF increased, compared with activated PBMCs cultured alone. Moreover, the release of IL-17AF was significantly reduced in transwell, compared with cell contacting conditions (Figure 5B,E), while the release of IFNγ was comparable in both systems (Figure 5A,D). The generation of classical Treg cells was also similarly up-regulated in the co-cultures allowing and preventing direct cell-to-cell contact (Figure 5C,F). No significant differences between the effects exerted by untreated and TI-treated ASCs, as well as between HD/ASCs and AS/ASCs were found. 

### 3.6. Comparison of AS/ASCs Effects in Allogeneic and Autologous System

The effect of AS/ASCs on IFNγ and IL-17AF secretion did not differ significantly between the co-cultures of these cells with PHA-activated allogeneic and autologous PBMCs, although it was less pronounced in the latter system, where only the trend toward down-regulation of IFNγ and up-regulation of IL-17AF was observed (Figure 6A,B). Additionally, the generation of classical Treg cells was similar when AS/ASCs were co-cultured with allogeneic and autologous target cells (Figure 6C). However, separately cultured PHA-activated PBMCs obtained from AS patients released significantly less anti-inflammatory IL-10 and TGFβ than respective allogeneic cells, and these differences existed even when AS/ASCs were added to the cultures (Figure 6D,E).

## 4. Discussion

Activated CD4^+^ T lymphocytes differentiate into several functionally specialized effector subsets. Between them, Th1, Th2, and Th17 cells contribute to the development of various autoimmune diseases, whereas Treg cells play a protective role [24,25]. In AS chronic activation of T lymphocytes leads to a quantitative imbalance between Th subsets and a significant increase in Th1/Th2 and Th17/Treg ratios [3,8,10,26,27]. Consistently, increased serum levels of the effector cytokines, IL-17, and IFNγ were reported [7,28]. These abnormalities are often associated with disease activity and normalize with clinical improvement [26,28,29]. The Th17 cells produce IL-17A and IL-17F, two structurally related members of the IL-17 family with overlapping pro-inflammatory activities that function as homo- and heterodimers (IL-17AF). Dysregulated expression of IL-17A and IL-17F is associated with AS and dual inhibition of these cytokines gives promising results [30]. Mesenchymal stem cells of various origins are known to support Treg cells and normalize the function of other Th subsets [14,25,31,32]. However, it is unknown whether AS/ASCs possess such capabilities. 

The generation of Th subsets is dependent on the master transcription factor activity [24,33]. Among them, T-bet is a Th1 lineage-defining transcription factor controlling IFNγ expression, while GATA3 activates the IL4/IL5/IL13 cytokine locus and directs Th2 cells formation [34]. The RORc, a master transcription factor of the Th17 lineage, coordinates transcription of IL-17A, whereas Foxp3 orchestrates Treg development and function, but represses the formation of other Th subsets [33,35,36]. Our present results show that HD/ASCs and AS/ASCs, co-cultured with activated CD4^+^ T cells (Figure 1) or PBMCs (Figure 2), have little effect on the expression of T-bet, but up-regulate the levels of GATA3, RORc, and FoxP3 mRNAs, causing significant decrease in T-bet/GATA3 and RORc/FoxP3 ratios in CD4^+^ T cells. Moreover, HD/ASCs and AS/ASCs, regardless of whether was cytokine (TI) primed or not, similarly modulated the expression of tested transcription factors. This is in contrast to other immunomodulatory functions of MSCs that are enhanced by “licensing” with pro-inflammatory cytokines [20,37], but consistent with other reports showing that ASCs and BM-MSCs up-regulate RORc mRNA expression independently of MSCs priming [38,39]. Thus, based on the above assessment, we conclude that both HD/ASCs and AS/ASCs can direct differentiation of Th cells towards domination of anti-inflammatory subsets. This shift, favorable in the context of AS pathology, was induced directly by ASCs and not changed by other cell types present among PBMCs. To further support this observation, the production of Th subset specific cytokines and the generation of Treg cells were evaluated. 

Consistently with up-regulation of RORc mRNA, a significant increase in IL-17AF concentration was found in the co-cultures of HD/ASCs and AS/ASCs with both types of target cells (Figure 3B and Figure 5B,E). By contrast, there was a discrepancy between unchanged T-bet mRNA level and IFNγ production which was either up-regulated in CD4^+^-ASCs co-cultures or reduced when ASCs were co-cultured with PBMCs (Figure 3A and Figure 5A,D). The reason for this likely lies in different concentrations of soluble factors that may influence IFNγ production. For example, PGE_2_ was reported to promote, in a dose-dependent manner, either Th1 or Th17 lineage commitment as well as to increase IFNγ^+^ cell proliferation and thus IFNγ production without altering T-bet expression [40]. The level of T-bet may also be critical because only CD4^+^ T cells with high T-bet expression mount protective IFNγ response [41]. Whatever the reason is, the present results of ASCs-PBMCs co-cultures confirmed the previously reported capability of MSCs to inhibit Th1/IFNγ generation [14,42,43,44,45]. In the co-cultures of ASCs with CD4^+^ T cells the divergence between FoxP3 mRNA expression and generation of classical (CD4^+^CD25^high^FoxP3^+^) Treg was found as well (Figure 1 and Figure 4). In these conditions, the number of Treg cells was decreased, despite up-regulation of FoxP3 mRNA level (Figure 1 and Figure 4A, respectively). By contrast, in ASCs-PBMCs co-cultures full compatibility between FoxP3 expression and Treg generation was noted (Figure 2 and Figure 4D). Our results are consistent with other data showing that human BM-MSCs can increase the formation of classical Treg only in the presence of monocytes/PBMCs, but fail to exert such effect when co-cultured with purified CD4^+^ T cells [46]. In the co-cultures with PBMCs, BM-MSCs were reported to act indirectly through skewing differentiation of monocytes towards anti-inflammatory type 2 macrophages that function as the accessory cells essential for Treg generation [46]. Our results supplement these observations by showing that the accessory cells, present in the PBMCs population, are critical to shifting ASCs-triggered T cell differentiation towards the anti-inflammatory direction, which is reflected by the up-regulation of classical Treg and down-regulation of Th1 cells. By contrast, direct action of ASCs seems to promote inflammatory response, mediated by Th1 and Th17 subsets and weakly controlled by classical Treg cells. In general, our results are compliant with numerous reports showing that in vitro and in vivo MSCs usually inhibit differentiation of Th1 lineage and favor classical Treg generation [14,42,43,44,45]. Unexpectedly, we found that ASCs up-regulate Th17 formation (Figure 1, Figure 2, Figure 3 and Figure 5). Data concerning MSCs’ impact on the generation of Th17 cells are controversial and both inhibitory [47,48,49,50,51,52] as well as promoting effects [42,43,44,45] were observed. Our results are consistent with the report demonstrating that human BM-MSCs can induce human Th17 and concomitantly suppress Th1 response via myeloid cell-mediated mechanism [45].

Besides classical Treg cells, a distinct population of regulatory cells, named Tr1 cells, exists [53,54]. The Tr1 cells produce predominantly IL-10 and TGFβ, and do not constitutively express FoxP3 [54]. Treg and Tr1 cells have different anatomical locations, control different types or stages of immune responses, but cooperate to maintain immune tolerance [53]. Because of this reason, we also assessed the production of Tr1-related anti-inflammatory TGFβ and IL-10 (Figure 4B–F). Contrary to expectation, the presence of ASCs did not increase these cytokines production but significantly decreased IL-10 concentrations in the co-cultures of HD/ASCs and AS/ASCs with PBMCs, and in the latter case, TGFβ level dropped as well. TGFβ was shown to be constitutively expressed and secreted by MSCs ([20,55,56] present results par. 3.4). Data concerning the effects of MSCs on TGFβ and IL-10 production are inconsistent. Neither MSCs nor MSCs conditioned medium were found to change TGFβ in target cells [31,55], but modulation of this cytokine in T cells was demonstrated [49]. As for IL-10, unchanged level [43], increased [46,47,48,50,55], or reduced [44,49] production of this cytokine in MSCs-PBMCs/T cells co-cultures were found. Although we did not evaluate the number of Tr1 cells, present results suggest that under the experimental design used in our study regulatory Tr1 cells are rather not generated. 

Transwell experiments showed that ASCs-induced formation of classical Treg cells and production of IFNγ by CD4^+^ T cells is mediated by soluble factors, whereas the production of IL-17AF is dependent on the direct cell-to-cell contact (Figure 5). Thus, we confirmed the observation of others concerning the production of IL-17 [48,51,57] and IFNγ [40] as well as the formation of Treg cells [46,55]. Regarding cell contact-dependent regulation of IL-17 by MSCs, the involvement of programmed death 1 and cyclooxygenase 2 pathways [48,51] were reported, while MSCs impact on Th1/IFNγ and Treg differentiation is known to be mediated by numerous soluble factors [13]. 

Interestingly, a comparison of AS/ASCs effects on allogeneic and autologous PBMCs revealed similarity of their action, i.e., the decrease in IFNγ secretion, an increase in IL-17AF release, and enhanced classical Treg formation (Figure 6A–C). Unfortunately, due to the small number of experiments performed there was rather a trend towards such modulation than statistically significant change. Despite this limitation, autologous PHA-activated PBMCs secreted significantly less IL-10 and TGFβ than allogeneic PBMCs (Figure 6D,E). Thus, PBMCs of AS patients have a defect in the production of these anti-inflammatory cytokines, which is not normalized by the presence of autologous ASCs. Similar impairment in IL-10 production by CD4^+^ T cells of AS patients was shown by others [9]. Although present results demonstrate the normal immunomodulatory impact of AS/ASCs on allogeneic and autologous T-cell differentiation, the inability of both HD/ASCs and AS/ASCs to down-regulate pro-inflammatory IL-17AF, and to up-regulate anti-inflammatory cytokines, weaken the possibility of therapeutic application of ASCs in ankylosing spondylitis.

There are three possible limitations in this study. The first is the relatively small size of the study group. It may affect the analysis and interpretation of the obtained results. Secondly, although the low surface expression of CD127 molecule allows more accurate identification of Treg cells [58], we did not check the expression of this molecule. Thus, the small contamination in the Treg subsets by activated T cell populations cannot be excluded. Finally, the question of the impact of AS patients’ treatment on tested ASCs activities remains open. Unfortunately, data on this problem are missing. It was only reported that treatment of rheumatic patients with glucocorticosteroids and anti-rheumatic DMARDs affected neither proliferation nor the clonogenic potential of their BM-MSCs [59], while NSAIDs exerted a dual effect on BM-MSCs proliferation in vitro, promoting it at low, but inhibiting at high concentration [60]. We suppose, however, that during a month-long expansion of AS/ASCs in vitro, required to obtain enough cell number to assess their immunomodulatory capability, drug-related effects disappeared.

## 5. Conclusions

We report that ASCs of AS patients and healthy donors exert comparable immunomodulatory impact on differentiation of allogeneic CD4^+^ T helper cells, and the influence of AS/ASCs on autologous cells is similar. This effect is mediated by soluble factors (down-regulation of IFNγ, Treg formation), and cell-to-cell contact (up-regulation of IL-17AF), and is not modified by ASCs licensing. By acting directly, ASCs shift differentiation of CD4^+^ T cells at transcriptional level towards anti-inflammatory subsets (decrease in Th1/Th2 and Th17/Treg ratios), but are not able to keep this drift at the functional level of the cells (down-regulate Treg cells, up-regulate IFNγ and IL-17AF). To maintain anti-inflammatory direction (up-regulation of Treg, suppression of IFNγ), ASCs require the assistance of the accessory cells. However, they are weak suppressors of pro-inflammatory Th17 response and do not enhance anti-inflammatory cytokine (TGFβ, IL-10) production. Therefore, the therapeutic usefulness of ASCs to AS patients remains uncertain. 

## Figures and Tables

**Figure 1 cells-10-00280-f001:**
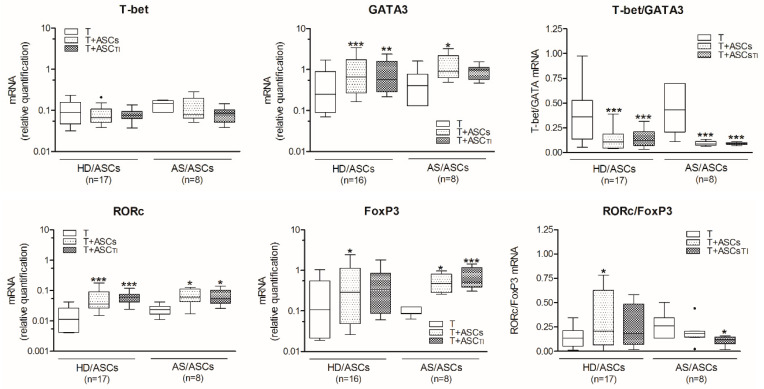
Direct effects of ASCs on the expression of Th subset specific transcription factors. T helper (CD3^+^CD4^+^) lymphocytes (T) were isolated from peripheral blood of 17 healthy donors, activated via CD3/CD28 pathway and co-cultured for 5 days with either untreated or TNF^+^IFNγ (TI) pre-stimulated 5 ASCs lines of healthy donors (HD/ASCs) or 8 ASCs lines obtained from AS patients (AS/ASCs). The ASCs-T cell co-cultures were performed using a random combination of both cell types and the number of experiments is shown (*n*). The levels of expression of mRNAs coding for T-bet, GATA3, RORc, and FoxP3 transcription factors, specific for Th1, Th2, Th17, and Treg cell subsets, respectively, were assessed by quantitative RT-PCR, as described in the Materials and methods. Results are expressed as the median (horizontal line) with interquartile range (IQR, box), lower and upper whiskers (data within 3/2xIQR), and outliers (points/dots) (Tukey’s box). One-way analysis of variance (ANOVA) with repeated measures and post-hoc Tukey test was used to evaluate the effect of ASCs (T vs. T + ASCs or ASCsTI; * *p* < 0.05; ** *p* < 0.001; *** *p* < 0.0001). The inter-group (HD vs. AS) differences, evaluated by the Mann-Whitney test, were statistically insignificant.

**Figure 2 cells-10-00280-f002:**
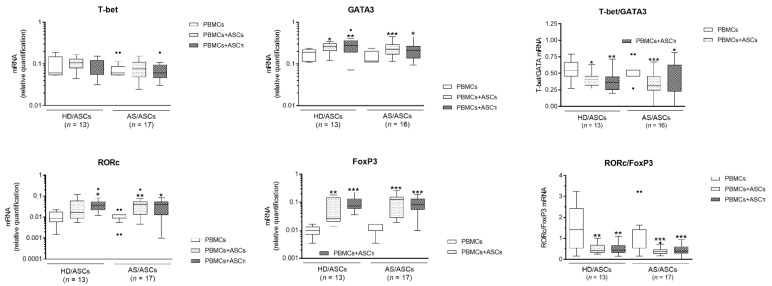
Changes in the expression of Th subset specific transcription factors in the co-cultures of ASCs with PBMCs. Explanations as in Figure 1, except that 5 HD/ASCs and 11 AS/ASCs lines were co-cultured with PHA-stimulated PBMCs that were isolated from peripheral blood of 13 healthy donors. The ASCs-PBMCs co-cultures were performed using a random combination of both cell types and the number of experiments is shown (*n*). Results are expressed as the median (horizontal line) with interquartile range (IQR, box), lower and upper whiskers (data within 3/2xIQR), and outliers (points/dots) (Tukey’s box). * *p* < 0.05; ** *p* < 0.001; *** *p* < 0.0001 for intra-group comparison (PBMCs vs. PBMCs + ASCs or ASCsTI); the inter-group differences were statistically insignificant.

**Figure 3 cells-10-00280-f003:**
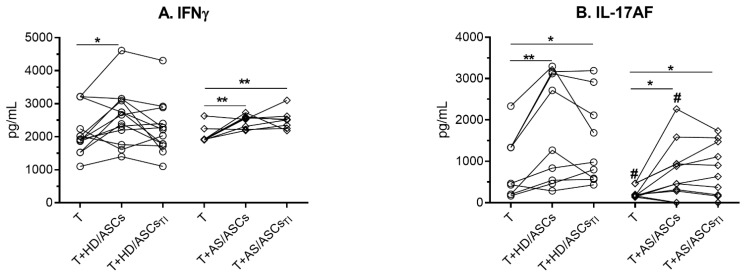
The effects of ASCs on the secretion of IFNγ (**A**) and IL-17AF (**B**). Cells were prepared and co-cultured as described in Figure 1. The concentrations of IFNγ and IL-17AF were measured in culture supernatants by specific ELISAs as described in the Material and methods. Lines between points identify cultures containing the same combination of ASCs and T cells. For intragroup comparisons (* *p* < 0.05; ** *p* < 0.001) data were analyzed by the one-way analysis of variance (ANOVA) with repeated measures and post-hoc Tukey test, while Mann–Whitney test (# *p* < 0.05) was used for inter-group comparison.

**Figure 4 cells-10-00280-f004:**
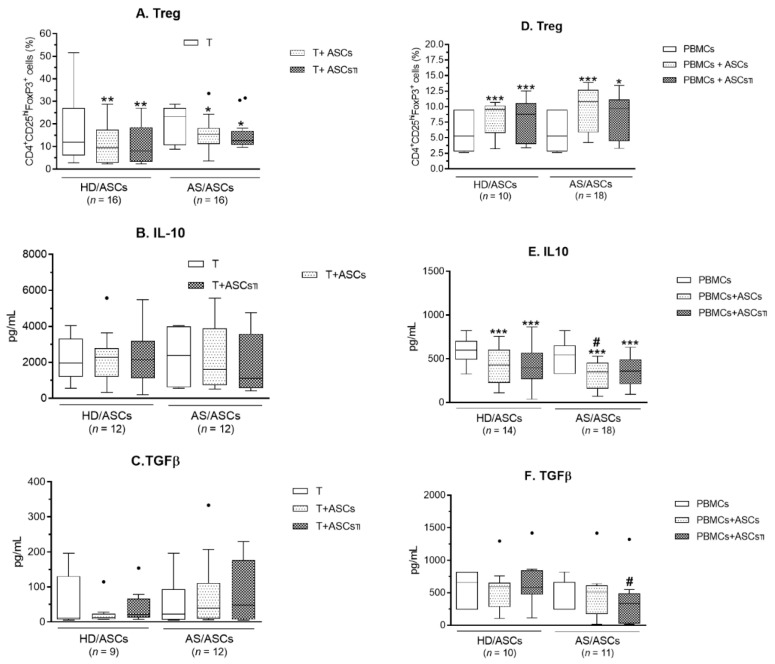
The effects of ASCs on Treg generation and secretion of anti-inflammatory cytokines. Cells were prepared and co-cultured as described in Figure 1 and Figure 2. For T cell-ASCs co-cultures, CD4^+^ T lymphocytes isolated from peripheral blood of 16 healthy volunteers were randomly combined with 5 HD/ASCs or 16 AS/ASCs lines, while for PBMCs-ASCs co-cultures the number of donors of PBMCs, HD/ASCs, and AS/ASCs was 20, 5, and 18, respectively. The number of Treg (**A**,**D**) cells was estimated and the concentrations of IL-10 (**B**,**E**) and TGFβ (**C**,**F**) in culture supernatants were measured by specific ELISAs as described in the Material and methods. Results are expressed as the median (horizontal line) with interquartile range (IQR, box), lower and upper whiskers (data within 3/2xIQR), and outliers (points/dots) (Tukey’s box); *n*—number of experiments performed. * *p* < 0.05; ** *p* < 0.001; *** *p* < 0.0001 for intra-group comparison. # *p* < 0.05 for inter-group comparison.

**Figure 5 cells-10-00280-f005:**
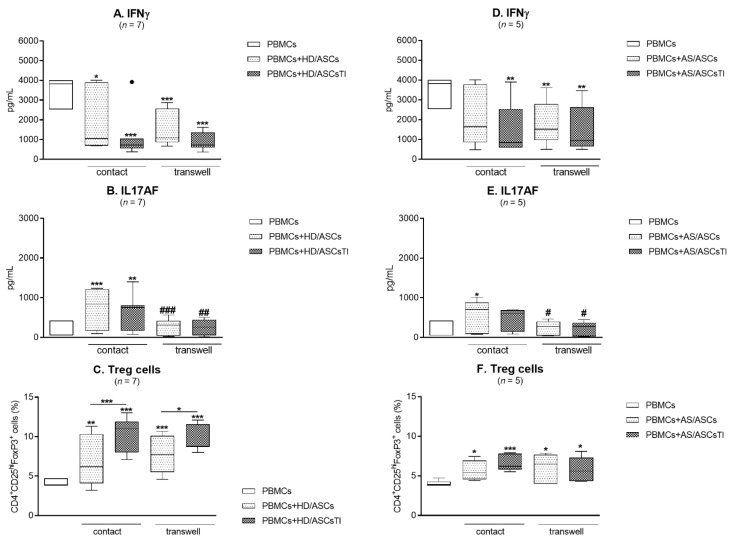
Generation of Treg cells and secretion of Th1, and Th17 specific cytokines in cell contacting and transwell co-cultures The number of Treg (**C**,**F**) cells was estimated and the concentrations of IFNγ (**A**,**D**) and IL-17AF (**B**,**E**) in culture supernatants were measured by specific ELISAs as described in the Material and methods. PBMCs, isolated from peripheral blood of 7 healthy volunteers were randomly combined with 5 HD/ASCs or 5 AS/ASCs lines, and the number of experiments performed (*n*) is shown. The co-cultures were done in conditions allowing direct cell-to-cell contact and in the transwell system, using the same combination of cells. Data are shown as the Tukey’s boxes. * *p* < 0.05; ** *p* < 0.001; *** *p* < 0.0001 for intra-group comparison. # *p* < 0.05, ## *p* < 0.001 ### *p* < 0. 0001 for inter-group comparison.

**Figure 6 cells-10-00280-f006:**
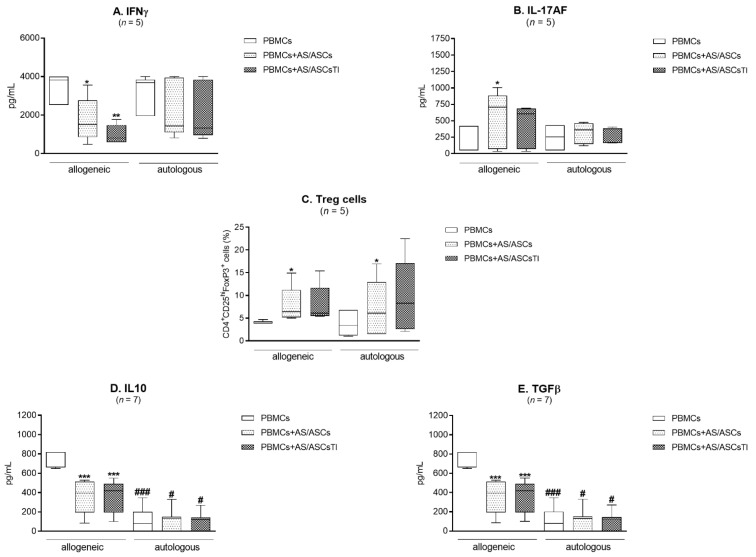
Comparison of AS/ASCs effects on allogeneic and autologous target cells. The concentrations of IFNγ (**A**), IL-17AF (**B**), IL-10 (**D**), TGFβ (**E**) in culture supernatants were measured by specific ELISAs and number of Treg (**C**) cells was estimated as described in the Material and methods. AS patients were co-cultured with autologous or allogeneic PBMCs obtained from 5–7 healthy volunteers. Data are shown as the Tukey’s boxes; *n*—number of experiments performed. * *p* < 0.05; ** *p* < 0.001; *** *p* < 0.0001 for intra-group comparison. # *p* < 0.05; ### *p* < 0.0001 for inter-group comparison.

**Table 1 cells-10-00280-t001:** Demographic and clinical characteristics of the patients.

Parameters	Ankylosing Spondylitis (AS) (*n* = 21)
Demographics	
Age, years	43.5 ± 2.79
Sex, female (F)/male (M), *n*	9 F/12 M
Disease duration, years	7.53 ± 0.98
Clinical data	
BASDAI, score	5.6 ± 0.48
ASDAS_CRP_, score	3.42 ± 0.22
BASFI, score	4.72 ± 0.61
BASMI, score	3.99 ± 0.43
HAQ, score	1.1 ± 0.17
Laboratory values	
CRP, mg/L	13.25 ± 3.38
ESR, mm/h	20.8 ± 4.13
Medications, %	
NSAIDs	90.0
Non-biologic DMARDs	25.0
Glucocorticosteroids	15.0

Except where indicated otherwise, values are the mean ± SEM. BASDAI, Bath Ankylosing Spondylitis Disease Activity Index; ASDAS, Ankylosing Spondylitis Disease Activity Score; BASFI, Bath Ankylosing Spondylitis Functional Index; BASMI, Bath Ankylosing Spondylitis Metrology Index; HAQ, Health Assessment Questionnaire; CRP, C-reactive protein; ESR, erythrocyte sedimentation rate; NSAIDs, non-steroid anti-inflammatory drugs; DMARDs, disease-modifying anti-rheumatic drugs.

## Data Availability

The data presented in this study are available in this article and its associated Appendix A (see above).

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
