# Peer review of "Modulatory Impact of Adipose-Derived Mesenchymal Stem Cells of Ankylosing Spondylitis Patients on T Helper Cell Differentiation"

_cells, 2021, doi:10.3390/cells10020280_

Round 1

Reviewer 1 Report

The manuscript is interesting and the data well presented, the significance of the results obtained is however limited by the low sample size = 21 patients.

-NSAIDs or corticosteroids treatments may affect cytokines levels and cell response to circulating cytokines. The authors should clarify if the pharmacological treatments could have affected their results.

-Some autoimmune disorders such as RA are gender-related, is there a gender effect in ankylosing spondylitis? 

Author Response

Dear Reviewer,

We would like to thank you for your valuable comments that allow us to improve our manuscript.

Sincerely,

Ewa Kuca-Warnawin

  1. The manuscript is interesting and the data well presented, the significance of the results obtained is however limited by the low sample size = 21 patients.

Due to ethical and clinical reasons, specimens of subcutaneous abdominal fat (approx.300 mg) were taken only from AS patients screened for amyloidosis who gave their written informed consent, and 18 G needle biopsy, not liposuction, was used - this information is added to the revised version of the manuscript (2.1). This limits both the size of the group of recruited patients and the number of obtainable ASCs. We have previously published several papers concerning ASCs obtained from a similar number of patients with various rheumatic diseases and the group size has never been questioned. Therefore, we believe that the number of patients included in the study is adequate to the recruitment possibilities and large enough to compare their function with ASCs from healthy donors.

  1. Kuca-Warnawin E, Janicka I, Szczęsny P, et al. Modulation of T-Cell Activation Markers Expression by the Adipose Tissue-Derived Mesenchymal Stem Cells of Patients with Rheumatic Diseases. Cell Transplant. 2020 Jan-Dec;29:963689720945682.
  2. Kuca-Warnawin E, Skalska U, Janicka I, et al. The Phenotype and Secretory Activity of Adipose-Derived Mesenchymal Stem Cells (ASCs) of Patients with Rheumatic Diseases. Cells. 2019 Dec 17;8(12):1659. doi: 10.3390/cells8121659.
  3. Skalska, U.; Kontny, E. Adipose-derived mesenchymal stem cells from infrapatellar FAT pad of patients with rheumatoid artrhritis and osteoarthritis have comparable immunomodulatory properties. Autoimmunity 2015, 49, 124-131.
  4. Skalska, U.; Kontny, E.; Prochorec-Sobieszek, M.; Maśliński, W. Intra-articular adipose-derived mesenchymal stem cells from rheumatoid arthritis patients maintain the function of chondrogenic differentiation. Rheumatology 2012, 51, 1757-1764.

  1. NSAIDs or corticosteroids treatments may affect cytokines levels and cell response to circulating cytokines. The authors should clarify if the pharmacological treatments could have affected their results.

There are scarce data on the impact of various drugs on MSCs activities. Treatment of rheumatic patients with glucocorticosteroids and anti-rheumatic DMARDs, such as methotrexate, leflunomide, or sulfasalazine, used in AS therapy, did not affect the clonogenic and proliferation capabilities of bone marrow-derived MSCs [1]. NSAIDs were reported to have a dual, dose-dependent effect on MSCs proliferation, promoting it at low (0.1 – 1 µM), while inhibiting at high (10-200 µM) concentrations [2]. However, data evaluating the impact of these drugs on the immunomodulatory capabilities of MSCs are missing. Therefore, it is hard to predict whether treatment of our AS patients exerted any effect on tested MSCs while inhibiting at high (10-200 µM) concentrations [2]. However, data evaluating the impact of these drugs on the immunomodulatory capabilities of MSCs are missing. Therefore, it is hard to predict whether treatment of our AS patients exerted any effect on tested MSCs activities. We believe, however, that during in vitro expansion of ASCs, which usually takes about one month, drug-related effects disappear. This assumption is supported by the observation that there is no significant difference in basic ASCs characterization between female and male AS patients (Table 3S), despite the more frequent use of glucocorticoids and DMARDs in females (Table 2S). This information is added to the Results section (3.1).

  1. Some autoimmune disorders such as RA are gender-related, is there a gender effect in ankylosing spondylitis?

Slightly more patients with AS are male than female, and the approximate male to female ratio is 2-3:1, but the sex distribution among patients with the earlier or milder disease is equal [3]. In our patients’ cohort there was no significant difference in clinical and demographic data between males and females, except the higher level of a systemic inflammation marker, CRP, in the female group, and a little more often the treatment of females with DMARDs and glucocorticosteroids (Table 2S). Moreover, the comparison of cytokine production and Treg generation in the co-cultures of target cells with ASCs obtained from male and female patients did not differ significantly (Table 3S). Therefore, in further analysis, the patients’ sex was not taken into account. This information is added to the Results section (3.1).

Tables comparing male and females patients are added in the Supplementary materials.

  1. Kastrianaki, MC.; Sidiropoulos, P.; Roche, S, et al. Functional, molecular and proteomic characterization of bone marrow mesenchymal stem cells in rheumatic arthritis. Ann Rheum Dis. 2008, 67, 741-749.
  2. Müller, M.; Raabe, O.; Addicks, K.; et al. Effects of non-steroidal anti-inflammatory drugs on proliferation, differentiation, and migration in equine mesenchymal stem cells. Cell Biol Int. 2011, 35, 235-248.
  3. Sieper, J.; Poddubny, D. Axial spondyloarthritis. Lancet 2017, 390, 73-84.

Please note that the references above were added to the manuscript’s reference list

Reviewer 2 Report

This paper investigates immunomodulatory properties of mesenchymal stem cells obtained from adipose tissue of AS patients (ASC) in view of the interest in using such cells therapeutically. The authors use rather complex co-culture systems - ASC from AS patients compared with ASC "lines" from healthy donors but supplied commercially (?similar passage numbers), and either activated or not; purified CD4+ cells (how pure?) activated by antiCD3/28, vs PBMC.  Perhaps inevitably the results that emerge are often not clear-cut. Reasonably convincingly the AS ASC seem to perform in the same way as the lines from health donors, and with no requirement for activation by TNF/IFN, and show increases in GATA3, RORc and Foxp3 expression. This is a mixed picture, with increased IL-17 - and therefore increased inflammation - together with improved Treg function. Such effects have been previously described, and seem very dependent on the co-culture system adopted. There are anomalies - Tbet expression was unchanged whereas IFN decreased in PBMC cultures. However, the amounts of IFN measured in these experiments are rather low and there is likely to be a technical explanation. Similarly, an increase in Foxp3 expression wasn't accompanied by increased phenotypic Treg. The authors speculate on both these points but don't have any data to clarify the findings. Lastly, there are minor points; there was no evidence of Tr1 induction; cell contact was not required for  most of the effects seen (except possibly the increase in IL-17, but if so for unknown reasons); autologous and allogeneic co-cultures were generally similar. In respect of this last point, the critical factor is not autologous and allogeneic - most of the co-culture systems used were allogeneic, but the ability of AS CD4+ T cells or PBMC to be modulated by either kind of ASC. 

The paper is much too long, with extensive introduction and discussion which cover general background material with which readers will be familiar.

Author Response

Dear Reviewer,

We would like to thank you for the valuable suggestions and comments that allow us to improve our manuscript, and for inspiring us to continue research on this subject.

Sincerely,

Ewa Kuca-Warnawin

  1. Cells between passages 3 and 5 were used for all experiments.
  2. The purity of the isolated lymphocytes was about 95%.
  3. In our opinion, it is hard to explain the decrease of IFNg in PBMC-ASCs co-cultures by technical reasons, because all concentration measurements from every experiment were done in duplicates and in the same assay plate. In addition, although in some experiments IFNg concentration declined from about 100 pg/ml to almost 0, in the other it fells from about 6 ng/ml to 500-1000 pg/ml [median (IQR) for: PBMCs vs PBMCs+HD/ASCsTI = 1342 (3753) vs 190 (686) p < 0.001; PBMCs vs AS/ASCsTI = 90 (2462) vs 3.9 (585), p < 0.05]. So, the amounts of IFNg in co-cultures were not low, but varied, and we noticed that the level of cytokine production in co-cultures is strongly related with particular PBMCs donor (Figure 3). Moreover, based on data of others we only speculated on discrepancy between the expression of transcription factor expression (T-bet, FoxP3) and the production of IFNg or generation of Treg cells. Obviously, we agree that we do not have any data to clarify the findings, but please note, that the aim of our work is only a comparison of AS/ASCs and HD/ASCs capability to modulate CD4+ T cell differentiation. As, in general, AS/ASCs and HD/ASCs do not differ in this respect, there is rather no need to explore the question more deeply, at least in the present paper.
  4. As the immunomodulatory function of AS/ASCs are little known, in the present paper we focused on the increase of the knowledge on this subject (papers evaluating other immunomodulatory functions of AS/ASCs are submitted for publication). We plan to investigate the reactivity of AS/PBMCs to both autologous and allogeneic ASCs more comprehensively, using several functional assays, but it is beyond the scope of this paper.
  5. Both, the Introduction and Discussion have been modified, shortened, and excessive general background information is removed, according to the reviewer’s suggestion.

Reviewer 3 Report

Review: “Modulatory impact of adipose-derived mesenchymal stem cells of ankylosing spondylitis patients on T helper cell differentiation”

Manuscript ID: cells-1044298

In this paper, the authors investigate the effect of ASCs (from HC or AS patients; w/o TNF and IFNG) on CD4 T-cell effector functions. In co-culture experiments they observed 1) an increase is Th2, Th17 and Treg transcriptional profile; 2) an up-regulation of pro-inflammatory cytokines IFNg and IL-17 on protein level; 3) that the observed effects are probably cell-contact independent and 4) that autologous and allogeneic cells have the same effect, except that autologous PBMCs already show a reduction of anti-inflammatory cytokines.

Overall, the study is well conducted, experiments seem well performed and the findings are of interest to the scientific community. Nevertheless, the manuscript would profit from some improvements:

Major points

  • The importance of the differences observed in experiments carried out in the presence of T-cells or total PBMCs is very difficult to delineate given that the cells were also treated differently. Therefore, the conclusion of indirect effects cannot really be made.
  • Are clinical data (treatment, disease activity, …) having an effect the ASCs in order to modulate T-cell responses?
  • The transwell experiments are of interest. Unfortunately, they were only performed in total PBMCs. Therefore, we do not know if cytokines produced by ASCs directly modulate T-cell responses. Also, it would have interesting to see which cytokine might regulate the effects, especially Treg upregulation. Blocking experiments of the suspected cytokines would have been great.
  • Given the partly occurring discrepancies between various experiments (e.g. FoxP3 and Treg), the transcriptomic analyses would profit from the involvement of further markers in order to enhance the confidence of the data shown. Although GATA3 for instance is the key transcription factor it might well be important for other cell subsets like Tregs.

Minor points

  • The authors only state TNF in the manuscript. If TNFa was used, the information should be visible.
  • Were IL-17A and F levels measured simultaneously? If yes, the authors should discuss this in the discussion section, given that they might have different effects.
  • Treg analysis would have benefitted from the inclusion of CD127 as a negative marker. Both CD25 and FoxP3 are known to be upregulated on T-cells upon stimulation.
  • The discussion section could be shortened and be more to the point of the study.

Author Response

Dear Reviewer,

We would like to thank the reviewer for valuable suggestions and comments that allow us to improve our manuscript, and for inspiring us to continue research on this subject.

Sincerely,

Ewa Kuca-Warnawin

Major points

  1. The importance of the differences observed in experiments carried out in the presence of T-cells or total PBMCs is very difficult to delineate given that the cells were also treated differently. Therefore, the conclusion of indirect effects cannot be made.

We agree that the conclusion of indirect effects in PBMCs-ASCs co-cultures is an oversimplification of the issue. Therefore, based on data of others, in the revised version of the manuscript, we suggest an only possible contribution of the accessory cells, present among PBMCs, to the effects exerted by ASCs – appropriate changes have been made throughout the manuscript. We used PHA to activate various types of cells in the PBMCs pool.

Anti-CD3 antibodies activate the TCR complex without antigenic peptide from the antigen-presenting cells. Anti-CD28 antibodies bind to CD28 and stimulate the T cells without CD80 or CD86 from antigen-presenting cells. Phytohaemagglutinin (PHA) is a lectin that has carbohydrate-binding specificity for a complex oligosaccharide containing galactose, N-acetylglucosamine, and mannose. Specifically, PHA binds to sugars on glycosylated surface proteins, including T cell receptor (TCR), and thereby crosslinks them. This triggers calcium-dependent signaling pathways leading to NFAT (nuclear factor of activated T cells) activation. Of course, the mechanism of action of PHA and CD3 / CD28 is not identical. However, PHA also works by triggering TCR (although not specific) so to some extent, the results from these two types of coculture can be comparable. Moreover, reports are showing that activation of T cells by CD3/CD28 and PHA produces very similar results [1]

  1. Jiao J et al. Comparison of two commonly used methods for stimulating T cells. Biotechnology Letters. Volume 41, pages1361–1371(2019)
  2. Are clinical data (treatment, disease activity, …) having an effect the ASCs in order to modulate T-cell responses?

 As for the possible effect of patients’ treatment and sex on ASCs activity, the relevant information has been added to the Results (3.10 and Supplementary materials of the revised manuscript version. We plan to analyze the association of ASCs characteristics with clinical data after collecting more information about the immunomodulatory capabilities of these cells. So far, data on this subject are scarce.

  1. 3. The transwell experiments are of interest. Unfortunately, they were only performed in total PBMCs. Therefore, we do not know if cytokines produced by ASCs directly modulate T-cell responses. Also, it would have interesting to see which cytokine might regulate the effects, especially Treg upregulation. Blocking experiments of the suspected cytokines would have been great.

Due to a limitation in fat sample collection, we were unable to perform transwell experiments using also CD4+ T cells as target cells. Because of the same reasons, it is impossible to run cytokine blocking experiments, suggested by the reviewer. We would like to explain that for ethical and clinical reasons, specimens of subcutaneous abdominal fat (approx.300 mg) were taken only from AS patients screened for amyloidosis who gave their written informed consent, and we used needle biopsy, not liposuction – this information is added to the revised version of the manuscript (2.2). This limits the number of obtainable ASCs, and it usually takes one month to expand ASCs to the number required to run experiments described in our manuscript. Moreover, it is known that several cytokines contribute to Treg (and other Th subsets) generation, so it would be difficult to block all of them, especially having a limited number of AS/ASCs. Also, we believe that it is not necessary for present work to perform such experiments, because AS/ASCs have similar activity as HD/ASCs. We would like to underline that our work aimed to compare the function of these cells while investigating the mechanism(s) is beyond the scope of the manuscript.

  1. Given the partly occurring discrepancies between various experiments (e.g. FoxP3 and Treg), the transcriptomic analyses would profit from the involvement of further markers to enhance the confidence of the data shown. Although GATA3 for instance is the key transcription factor it might well be important for other cell subsets like Tregs.

Because of the limitations described above, we investigated only the well-established lineage-defining transcription factors. In our opinion, the discrepancies between transcription factors expression and Treg generation/Th subset specific cytokine production, observed primarily in CD4+ T-ASCs co-cultures, inspire us to continue research on the mechanism(s) used by ASCs of healthy donors to modulate Th differentiation, and the contribution of the accessory cells in this process, but it requires another experimental approach.

Minor points

  1. The authors only state TNF in the manuscript. If TNFa was used, the information should be visible.

The choice of the name TNF / TNFalpha is still controversial. In 1975, TNF was identified in human serum. Functional and sequential homology of TNF and the previously discovered cytotoxic factor lymphotoxin was reported and resulted in renaming of TNF as TNF-α and lymphotoxin as TNF-β [2] a close homologue of lymphotoxin, the lymphotoxin-β, was discovered in 1999. Subsequently, at the Seventh International TNF Congress (May 17-21, 1998; Hyannis, Massachusetts), the name “TNF-β” was changed to “lymphotoxin-α.” Concurrently, “TNF-α” became an orphan term with no meaning different from the original term, “TNF,” which was reinstated as the official name of the cytokine [ref].  For this reason, it seems appropriate to use the name TNF [1].

  1. Grimstad Ø. Tumor Necrosis Factor and the Tenacious α. JAMA Dermatology May 2016 Volume 152, Number 5, 5572. I
  2. Were IL-17A and F levels measured simultaneously? If yes, the authors should discuss this in the discussion section, given that they might have different effects.

We have measured the concentration of IL-17AF heterodimer. We have tried to make this information very clear throughout the manuscript.

Th17 cells can produce proinflammatory IL-17A and IL-17 F, where IL-17A is considered more potent than IL-17 F. These two isoforms create dimers: in body fluids, homodimers IL-17AA and IL-17FF and also heterodimer IL-17AF could be detected [1]. However, our previous research showed that IL-17AF heterodimer is the most frequent form of IL-17 present in blood and bone marrow [2]. Therefore, in our current work, we focused on the assessment of the effect of ASC on IL-17AF concentration

  1. Wright JF, Guo YJ, Quazi A, Luxenberg DP, et al. Identification of an interleukin 17F/17A heterodimer in activated human CD4+ T cells. J Biol Chem. 2007;282(18):13447–55
  2. Kuca-Warnawin E, Kurowska W, Prochorec-Sobieszek M, et al. Rheumatoid arthritis bone marrow environment supports Th17 response. Arthritis Res Ther. 2017;19(1):274. Published 2017 Dec 8. doi:10.1186/s13075-017-1483-x
  3. 3. Treg analysis would have benefitted from the inclusion of CD127 as a negative marker. Both CD25 and FoxP3 are known to be upregulated on T-cells upon stimulation

We agree that the inclusion of CD127 in the analysis would be beneficial. Unfortunately, it is impossible to include another marker for analysis after data acquiring.

Please note that in our study we evaluated the change of percentage of regulatory T cells concerning the control in which the cells were activated. So, all the CD4 and PBMC variants we have studied were activated. Moreover, our previous work has shown that co-culture of ASC with PBMC reduces the expression of CD25 on T cells [1]. For this reason, we believe that in our experimental model we see an increase in the percentage of regulatory cells.

  1. Kuca-Warnawin E, Janicka I, Szczęsny P, et al. Modulation of T-Cell Activation Markers Expression by the Adipose Tissue-Derived Mesenchymal Stem Cells of Patients with Rheumatic Diseases. Cell Transplant. 2020 Jan-Dec;29:963.
  2. The discussion section could be shortened and be more to the point of the study.

Both, the Introduction and Discussion have been modified, shortened, and excessive general background information is removed, according to reviewer’s suggestion.

Round 2

Reviewer 1 Report

Despite the manuscript has been certainly improved still some methodological concerns may limit the significance of the work done, the sample size is low and the population is heterogeneous with several confounding factors. 

I suggest clarifying in a specific paragraph in the discussion section the limitation of this work.

Author Response

Dear Reviewer,

We would like to thank you for your valuable comments that allow us to improve our manuscript.

Sincerely,

Authors,

  1. Despite the manuscript has been certainly improved still some methodological concerns may limit the significance of the work done, the sample size is low and the population is heterogeneous with several confounding factors. I suggest clarifying in a specific paragraph in the discussion section the limitation of this work.

As suggested by the reviewer, a paragraph on the limitations of our study has been added to the “Discussion” section.

Reviewer 2 Report

The authors have responded appropriately to reviewers' comments as best they can and the manuscript is now significantly improved. However, I think that there should be three changes, in descending order of importance. These are:

  1. The data on IFN production shown in Fig 3C are simply not believable, despite the authors'comments which I appreciate; they are alos completrely inconsistent with all the other, perfectly acceptable, data on IFN production as seen in Fig 3A, Fig 5 and Fig 6.  Therefore I think the data on PBMC in Fig 3 should be removed; this gets rid of an anomaly in the results - the difference between CD4+ cells and PBMC which doesn't have any obvious explanation, or indeed implications for our understanding. It will also allow conventional plotting of IFN on a linear rather than log2 scale in Fig 3A and B.
  2. I don't think the small amount of data on allogeneic vs autologous MSC are worth including. I can't see why differences would be expected, or what they would mean if they were found, unlike the differences between AS patients and healthy donors.
  3. The authors have responded to comments about drug treatment and other issues on patient characteristics, but this is in the Results section together with references. The comments need to be in the Discussion and removed from results - lines 145-151.

Author Response

please check the attached for response to reviewer 2' comments

Reviewer 3 Report

The authors state that additional experiments are not possible due to 'ethical and clinical' reasons. That is very unfortunate because the manuscript would have benefitted greatly from these data.

In regard to the effects of treatment, etc. on ASCs the authors mention that these effects might be lost during the expansion phase that is needed for the experimental setup. Still, I feel that the authors should analyze the existing data in this regard in order to rule out any unexpected effects.

Also, the authors should add the information on missing CD127 as a negative marker for Tregs in the limitations section of their discussion.

Author Response

Please check the attached file for the response to reviewer 3'commnets

This manuscript is a resubmission of an earlier submission. The following is a list of the peer review reports and author responses from that submission.